# A State-Space Model for Decoding Auditory Attentional Modulation from MEG in a Competing-Speaker Environment

**Sahar Akram**[1,2]**, Jonathan Z. Simon**[1,2,3]**, Shihab Shamma**[1,2]**, and Behtash Babadi**[1,2]
[1] Department of Electrical and Computer Engineering,
[2] Institute for Systems Research, [3] Department of Biology
University of Maryland
College Park, MD 20742, USA
{sakram,jzsimon,sas,behtash}@umd.edu

## Abstract

Humans are able to segregate auditory objects in a complex acoustic scene, through an interplay of bottom-up feature extraction and top-down selective attention in the brain. The detailed mechanism underlying this process is largely unknown and the ability to mimic this procedure is an important problem in artificial intelligence and computational neuroscience. We consider the problem of decoding the attentional state of a listener in a competing-speaker environment from magnetoencephalographic (MEG) recordings from the human brain. We develop a behaviorally inspired state-space model to account for the modulation of the MEG with respect to attentional state of the listener. We construct a decoder based on the maximum *a posteriori* (MAP) estimate of the state parameters via the Expectation-Maximization (EM) algorithm. The resulting decoder is able to track the attentional modulation of the listener with multi-second resolution using only the envelopes of the two speech streams as covariates. We present simulation studies as well as application to real MEG data from two human subjects. Our results reveal that the proposed decoder provides substantial gains in terms of temporal resolution, complexity, and decoding accuracy.

## 1 Introduction

Segregating a speaker of interest in a multi-speaker environment is an effortless task we routinely perform. It has been hypothesized that after entering the auditory system, the complex auditory signal resulted from concurrent sound sources in a crowded environment is decomposed into acoustic features. An appropriate binding of the relevant features, and discounting of others, leads to forming the percept of an auditory object [1, 2, 3]. The complexity of this process becomes tangible when one tries to synthesize the underlying mechanism known as the cocktail party problem [4, 5, 6, 7].

In a number of recent studies it has been shown that concurrent auditory objects even with highly overlapping spectrotemporal features, are neurally encoded as a distinct object in auditory cortex and emerge as fundamental representational units for high-level cognitive processing [8, 9, 10]. In the case of listening to speech, it has recently been demonstrated by Ding and Simon [8], that the auditory response manifested in MEG is strongly modulated by the spectrotemporal features of the speech. In the presence of two speakers, this modulation appears to be strongly correlated with the temporal features of the attended speaker as opposed to the unattended speaker (See Figure 1–A). Previous studies employ time-averaging across multiple trials in order to decode the attentional state of the listener from MEG observations. This method is only valid when the subject is attending to a single speaker during the entire trial. In a real-world scenario, the attention of the listener can switch dynamically from one speaker to another. Decoding the attentional target in this scenario requires a

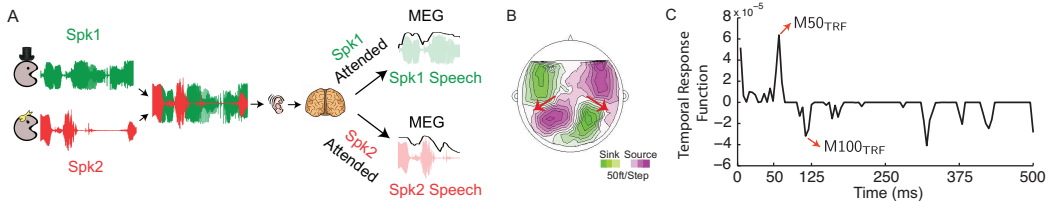

Figure 1: A) Schematic depiction of auditory object encoding in the auditory cortex. B) The MEG magnetic field distribution of the first DSS component shows a stereotypical pattern of neural activity originating separately in the left and right auditory cortices. Purple and green contours represent the magnetic field strength. Red arrows schematically represent the locations of the dipole currents, generating the measured magnetic field. C) An example of the TRF, estimated from real MEG data. Significant TRF components analogous to the well-known M50 and M100 auditory responses are marked in the plot.

dynamic estimation framework with high temporal resolution. Moreover, the current techniques use the full spectrotemporal features of the speech for decoding. It is not clear whether the decoding can be carried out with a more parsimonious set of spectrotemporal features.

In this paper, we develop a behaviorally inspired state-space model to account for the modulation of MEG with respect to the attentional state of the listener in a double-speaker environment. MAP estimation of the state-space parameters given MEG observations is carried out via the EM algorithm. We present simulation studies as well as application to experimentally acquired MEG data, which reveal that the proposed decoder is able to accurately track the attentional state of a listener in a double-speaker environment while selectively listening to one of the two speakers. Our method has three main advantages over existing techniques. First, the decoder provides estimates with sub-second temporal resolution. Second, it only uses the envelopes of the two speech streams as the covariates, which is a substantial reduction in the dimension of the spectrotemporal feature set used for decoding. Third, the principled statistical framework used in constructing the decoder allows us to obtain confidence bounds on the estimated attentional state.

The paper is organized as follows. In Section 2, we introduce the state-space model and the proposed decoding algorithm. We present simulation studies to test the decoder in terms of robustness with respect to noise as well as tracking performance and apply to real MEG data recorded from two human subjects in Section 3. Finally, we discuss the future directions and generalizations of our proposed framework in Section 4.

## 2   Methods

We first consider the forward problem of relating the MEG observations to the spectrotemporal features of the attended and unattended speech streams. Next, we consider the inverse problem where we seek to decode the attentional state of the listener given the MEG observations and the temporal features of the two speech streams.

### 2.1   The Forward Problem: Estimating the Temporal Response Function

Consider a task where the subject is passively listening to a speech stream. Let the discrete-time MEG observation at time $t$, sensor $j$, and trial $r$ be denoted by $x_{t,j,r}$, for $t = 1, 2, \cdots, T$, $j = 1, 2, \cdots, M$ and $r = 1, 2, \cdots, R$. The stimulus-irrelevant neural activity can be removed using denoising source separation (DSS) [11]. The DSS algorithm is a blind source separation method that decomposes the data into $T$ temporally uncorrelated components by enhancing consistent components over trials and suppressing noise-like components of the data, with no knowledge of the stimulus or timing of the task. Let the time series $y_{1,r}, y_{2,r}, \cdots, y_{T,r}$ denote the first significant component of the DSS decomposition, denoted hereafter by MEG data. In an auditory task, this component has a field map which is consistent with the stereotypical auditory response in MEG (See Figure 1–B). Also, let $E_t$ be the speech envelope of the speaker at time $t$ in dB scale. In a linear model, the MEG data is linearly related to the envelope of speech as:

$$y_{t,r} = \tau_t * E_t + v_{t,r}, \tag{1}$$

where $\tau_t$ is a linear filter of length $L$ denoted by temporal response function (TRF), $*$ denotes the convolution operator, and $v_{t,r}$ is a nuisance component accounting for trial-dependent and stimulus-independent components manifested in the MEG data. It is known that the TRF is a sparse filter, with significant components analogous to the M50 and M100 auditory responses ([9, 8], See Figure 1–C). A commonly-used technique for estimating the TRF is known as Boosting ([12, 9]), where the components of the TRF are greedily selected to decrease the mean square error (MSE) of the fit to the MEG data. We employ an alternative estimation framework based on $\ell_1$-regularization. Let $\boldsymbol{\tau} := [\tau_L, \tau_{L-1}, \cdots, \tau_1]'$ be the time-reversed version of the TRF filter in vector form, and let $\mathbf{E}_t := [E_t, E_{t-1}, \cdots, E_{t-L+1}]'$. In order to obtain a sparse estimate of the TRF, we seek the $\ell_1$-regularized estimate:

$$\widehat{\boldsymbol{\tau}} = \underset{\boldsymbol{\tau}}{\operatorname{argmin}} \sum_{r,t=1}^{R,T} \|y_{t,r} - \boldsymbol{\tau}'\mathbf{E}_t\|_2^2 + \gamma\|\boldsymbol{\tau}\|_1, \tag{2}$$

where $\gamma$ is the regularization parameter. The above problem can be solved using standard optimization software. We have used a fast solver based on iteratively re-weighted least squares [13]. The parameter $\gamma$ is chosen by two-fold cross-validation, where the first half of the data is used for estimating $\boldsymbol{\tau}$ and the second half is used to evaluate the goodness-of-fit in the MSE sense. An example of the estimated TRF is shown in Figure 1–C. In a competing-speaker environment, where the subject is only attending to one of the two speakers, the linear model takes the form:

$$y_{t,r} = \tau_t^a * E_t^a + \tau_t^u * E_t^u + v_{t,r}, \tag{3}$$

with $\tau_t^a$, $E_t^a$, $\tau_t^u$, and $E_t^u$, denoting the TRF and envelope of the attended and unattended speakers, respectively. The above estimation framework can be generalized to the two-speaker case by replacing the regressor $\boldsymbol{\tau}'\mathbf{E}_t$ with $\boldsymbol{\tau}^{a'}\mathbf{E}_t^a + \boldsymbol{\tau}^{u'}\mathbf{E}_t^u$, where $\boldsymbol{\tau}^a$, $\mathbf{E}_t^a$, $\boldsymbol{\tau}^u$, and $\mathbf{E}_t^u$ are defined in a fashion similar to the single-speaker case. Similarly, the regularization $\gamma\|\boldsymbol{\tau}\|_1$ is replaced by $\gamma^a\|\boldsymbol{\tau}^a\|_1 + \gamma^u\|\boldsymbol{\tau}^u\|_1$.

## 2.2 The Inverse Problem: Decoding Attentional Modulation

### 2.2.1 Observation Model

Let $y_{1,r}, y_{2,r}, \cdots, y_{T,r}$ denote the MEG data time series at trial $r$, for $r = 1, 2, \cdots, R$ during an observation period of length $T$. For a window length $W$, let

$$\mathbf{y}_{k,r} := \left[ y_{(k-1)W+1,r}, y_{(k-1)W+2,r}, \cdots, y_{kW,r} \right], \tag{4}$$

for $k = 1, 2, \cdots, K := \lfloor T/W \rfloor$. Also, let $E_{i,t}$ be the speech envelope of speaker $i$ at time $t$ in dB scale, $i = 1, 2$. Let $\tau_t^a$ and $\tau_t^u$ denote the TRFs of the attended and unattended speakers, respectively. The MEG predictors in the linear model are given by:

$$\begin{cases} e_{1,t} := \tau_t^a * E_{1,t} + \tau_t^u * E_{2,t} & \text{attending to speaker 1} \\ e_{2,t} := \tau_t^a * E_{2,t} + \tau_t^u * E_{1,t} & \text{attending to speaker 2} \end{cases}, \quad t = 1, 2, \cdots, T. \tag{5}$$

Let

$$\mathbf{e}_{i,k} := \left[ e_{i,(k-1)W+1}, e_{i,(k-1)W+2}, \cdots, e_{i,kW} \right], \quad \text{for } i = 1, 2 \text{ and } k = 1, 2, \cdots, K. \tag{6}$$

Recent work by Ding and Simon [8] suggests that the MEG data $\mathbf{y}_k$ is more correlated with the predictor $\mathbf{e}_{i,k}$ when the subject is attending to the $i$th speaker at window $k$. Let

$$\theta_{i,k,r} := \arccos\left( \left\langle \frac{\mathbf{y}_{k,r}}{\|\mathbf{y}_{k,r}\|_2}, \frac{\mathbf{e}_{i,k}}{\|\mathbf{e}_{i,k}\|_2} \right\rangle \right) \tag{7}$$

denote the empirical correlation between the observed MEG data and the model prediction when attending to speaker $i$ at window $k$ and trial $r$. When $\theta_{i,k,r}$ is close to 0 ($\pi$), the MEG data and its predicted value are highly (poorly) correlated. Inspired by the findings of Ding and Simon [8], we model the statistics of $\theta_{i,k,r}$ by the von Mises distribution [14]:

$$p(\theta_{i,k,r}) = \frac{1}{\pi I_0(\kappa_i)} \exp(\kappa_i \cos(\theta_{i,k,r})), \qquad \theta_{i,k,r} \in [0, \pi], \quad i = 1, 2 \tag{8}$$

where $I_0(\cdot)$ is the zeroth order modified Bessel function of the first kind, and $\kappa_i$ denotes the spread parameter of the von Mises distribution for $i = 1, 2$. The von Mises distribution gives more (less) weight to higher (lower) values of correlation between the MEG data and its predictor and is pretty robust to gain fluctuations of the neural data. The spread parameter $\kappa_i$ accounts for the concentration of $\theta_{i,k,r}$ around 0. We assume a conjugate prior of the form $p(\kappa_i) \propto \frac{\exp(c_0 d\kappa_i)}{I_0(\kappa_i)^d}$ over $\kappa_i$, for some hyper-parameters $c_0$ and $d$.

### 2.2.2 State Model

Suppose that at each window of observation, the subject is attending to either of the two speakers. Let $n_{k,r}$ be a binary variable denoting the attention state of the subject at window $k$ and trial $r$:

$$n_{k,r} = \begin{cases} 1 & \text{attending to speaker 1} \\ 0 & \text{attending to speaker 2} \end{cases} \tag{9}$$

The subjective experience of attending to a specific speech stream among a number of competing speeches reveals that the attention often switches to the competing speakers, although not intended by the listener. Therefore, we model the statistics of $n_{k,r}$ by a Bernoulli process with a success probability of $q_k$:

$$p(n_{k,r}|q_k) = q_k^{n_{k,r}}(1 - q_k)^{1-n_{k,r}}. \tag{10}$$

A value of $q_k$ close to 1 (0) implies attention to speaker 1 (2). The process $\{q_k\}_{k=1}^K$ is assumed to be common among different trials. In order to model the dynamics of $q_k$, we define a variable $z_k$ such that

$$q_k = \mathsf{logit}^{-1}(z_k) := \frac{\exp(z_k)}{1 + \exp(z_k)}. \tag{11}$$

When $z_k$ tends to $+\infty$ ($-\infty$), $q_k$ tends to 1 (0). We assume that $z_k$ obeys AR(1) dynamics of the form:

$$z_k = z_{k-1} + w_k, \tag{12}$$

where $w_k$ is a zero-mean i.i.d. Gaussian random variable with a variance of $\eta_k$. We further assume that $\eta_k$ are distributed according to the conjugate prior given by the inverse-Gamma distribution with hyper-parameters $\alpha$ (shape) and $\beta$ (scale).

### 2.2.3 Parameter Estimation

Let

$$\Omega := \left\{ \kappa_1, \kappa_2, \{z_k\}_{k=1}^K, \{\eta_k\}_{k=1}^K \right\} \tag{13}$$

be the set of parameters. The log-posterior of the parameter set $\Omega$ given the observed data $\{\theta_{i,k,r}\}_{i,k,r=1}^{2,T,R}$ is given by:

$$\log p\left(\Omega \Big| \{\theta_{i,k,r}\}_{i,k,r=1}^{2,K,R}\right) = \sum_{r,k=1}^{R,K} \log\left[\frac{q_k}{\pi I_0(\kappa_1)} \exp\left(\kappa_1 \cos(\theta_{1,k,r})\right) + \frac{1-q_k}{\pi I_0(\kappa_2)} \exp\left(\kappa_2 \cos(\theta_{2,k,r})\right)\right]$$
$$+ \left[(\kappa_1 + \kappa_2)c_0 d - d\left(\log I_0(\kappa_1) + \log I_0(\kappa_2)\right)\right]$$
$$- \sum_{r,k=1}^{R,K} \left\{\frac{1}{2\eta_k}(z_k - z_{k-1})^2 + \frac{1}{2}\log\eta_k + (\alpha+1)\log\eta_k + \frac{\beta}{\eta_k}\right\} + \mathsf{cst}.$$

where $\mathsf{cst}.$ denotes terms that are not functions of $\Omega$. The MAP estimate of the parameters is difficult to obtain given the involved functional form of the log-posterior. However, the complete data log-posterior, where the unobservable sequence $\{n_{k,r}\}_{k=1,r=1}^{K,R}$ is given, takes the form:

$$\log p\left(\Omega \Big| \{\theta_{i,k,r}, n_{k,r}\}_{i,k,r=1}^{2,K,R}\right) = \sum_{r,k=1}^{R,K} n_{k,r} \left[\kappa_1 \cos(\theta_{1,k,r}) - \log I_0(\kappa_1) + \log q_k\right]$$
$$+ \sum_{r,k=1}^{R,K} (1-n_{k,r})\left[\kappa_2 \cos(\theta_{2,k,r}) - \log I_0(\kappa_2) + \log(1-q_k)\right]$$
$$+ \left[(\kappa_1 + \kappa_2)c_0 d - d\left(\log I_0(\kappa_1) + \log I_0(\kappa_2)\right)\right]$$
$$- \sum_{r,k=1}^{R,K} \left\{\frac{1}{2\eta_k}(z_k - z_{k-1})^2 + \frac{1}{2}\log\eta_k + (\alpha+1)\log\eta_k + \frac{\beta}{\eta_k}\right\} + \mathsf{cst}.$$

The log-posterior of the parameters given the complete data has a tractable functional form for optimization purposes. Therefore, by taking $\{n_{k,r}\}_{k=1,r=1}^{K,R}$ as the unobserved data, we can estimate

$\Omega$ via the EM algorithm [15]. Using Bayes' rule, the expectation of $n_{k,r}$, given $\{\theta_{i,k,r}\}_{i,k,r=1}^{2,K,R}$ and current estimates of the parameters $\Omega^{(\ell)} := \{\kappa_1^{(\ell)}, \kappa_2^{(\ell)}, \{z_k^{(\ell)}\}_{k=1}^K, \{\eta_k^{(\ell)}\}_{k=1}^K\}$ is given by:

$$\mathbb{E}\left\{n_{k,r}\Big|\{\theta_{i,k,r}\}_{i,k,r=1}^{2,K,R}, \Omega^{(\ell)}\right\} = \frac{\frac{q_k^{(\ell)}}{\pi I_0\left(\kappa_1^{(\ell)}\right)}\exp\left(\kappa_1^{(\ell)}\cos\left(\theta_{1,k,r}\right)\right)}{\frac{q_k^{(\ell)}}{\pi I_0\left(\kappa_1^{(\ell)}\right)}\exp\left(\kappa_1^{(\ell)}\cos\left(\theta_{1,k,r}\right)\right) + \frac{1-q_k^{(\ell)}}{\pi I_0\left(\kappa_2^{(\ell)}\right)}\exp\left(\kappa_2^{(\ell)}\cos\left(\theta_{2,k,r}\right)\right)}.$$

Denoting the expectation above by the shorthand $\mathbb{E}^{(\ell)}\{n_{k,r}\}$, the M-step of the EM algorithm for $\kappa_1^{(\ell+1)}$ and $\kappa_2^{(\ell+1)}$ gives:

$$\kappa_i^{(\ell+1)} = A^{-1}\left(\frac{c_0 d + \sum_{r,k=1}^{R,K}\varepsilon_{i,k,r}^{(\ell)}\cos\left(\theta_{i,k,r}\right)}{d + \sum_{r,k=1}^{R,K}\varepsilon_{i,k,r}^{(\ell)}}\right), \quad \varepsilon_{i,k,r}^{(\ell)} = \begin{cases} \mathbb{E}^{(\ell)}\{n_{k,r}\} & i=1 \\ 1 - \mathbb{E}^{(\ell)}\{n_{k,r}\} & i=2 \end{cases}, \quad (14)$$

where $A(x) := I_1(x)/I_0(x)$, with $I_1(\cdot)$ denoting the first order modified Bessel function of the first kind. Inversion of $A(\cdot)$ can be carried out numerically in order to find $\kappa_1^{(\ell+1)}$ and $\kappa_2^{(\ell+1)}$. The M-step for $\{\eta_k\}_{k=1}^K$ and $\{z_k\}_{k=1}^K$ corresponds to the following maximization problem:

$$\underset{\{z_k,\eta_k\}_{k=1}^K}{\arg\max}\sum_{r,k=1}^{R,K}\left[\mathbb{E}^{(\ell)}\{n_{k,r}\}z_k - \log(1+\exp(z_k)) - \frac{1}{2\eta_k}\left[(z_k-z_{k-1})^2+2\beta\right] - \frac{1+2(\alpha+1)}{2}\log\eta_k\right].$$

An efficient approximate solution to this maximization problem is given by another EM algorithm, where the E-step is the point process smoothing algorithm [16, 17] and the M-step updates the state variance sequence [18]. At iteration $m$, given an estimate of $\eta_k^{(\ell+1)}$, denoted by $\eta_k^{(\ell+1,m)}$, the forward pass of the E-step for $k = 1, 2, \cdots, K$ is given by:

$$\begin{cases} \bar{z}_{k|k-1}^{(\ell+1,m)} = \bar{z}_{k-1|k-1}^{(\ell+1,m)} \\ \sigma_{k|k-1}^{(\ell+1,m)} = \sigma_{k-1|k-1}^{(\ell+1,m)} + \frac{\eta_k^{(\ell+1,m)}}{R} \\ \bar{z}_{k|k}^{(\ell+1,m)} = \bar{z}_{k|k-1}^{(\ell+1,m)} + \sigma_{k|k-1}^{(\ell+1,m)}\left[\sum_{r=1}^R \mathbb{E}^{(\ell)}\{n_{k,r}\} - R\frac{\exp\left(\bar{z}_{k|k}^{(\ell+1,m)}\right)}{1+\exp\left(\bar{z}_{k|k}^{(\ell+1,m)}\right)}\right] \\ \sigma_{k|k}^{(\ell+1,m)} = \left[\frac{1}{\sigma_{k|k-1}^{(\ell+1,m)}} + R\frac{\exp\left(\bar{z}_{k|k}^{(\ell+1,m)}\right)}{\left(1+\exp\left(\bar{z}_{k|k}^{(\ell+1,m)}\right)\right)^2}\right]^{-1} \end{cases} \quad (15)$$

and for $k = K-1, K-2, \cdots, 1$, the backward pass of the E-step is given by:

$$\begin{cases} s_k^{(\ell+1,m)} = \sigma_{k|k}^{(\ell+1,m)}/\sigma_{k+1|k}^{(\ell+1,m)} \\ \bar{z}_{k|K}^{(\ell+1,m)} = \bar{z}_{k|k}^{(\ell+1,m)} + s_k^{(\ell+1,m)}\left(\bar{z}_{k+1|K}^{(\ell+1,m)} - \bar{z}_{k+1|k}^{(\ell+1,m)}\right) \\ \sigma_{k|K}^{(\ell+1,m)} = \sigma_{k|k}^{(\ell+1,m)} + s_k^{(\ell+1,m)}\left(\sigma_{k+1|K}^{(\ell+1,m)} - \sigma_{k+1|k}^{(\ell+1,m)}\right)s_k^{(\ell+1,m)} \end{cases} \quad (16)$$

Note that the third equation in the forward filter is non-linear in $\bar{z}_{k|k}^{(\ell+1,m)}$, and can be solved using standard techniques (e.g., Newton's method). The M-step gives the updated value of $\eta_k^{(\ell+1,m+1)}$ as:

$$\eta_k^{(\ell+1,m+1)} = \frac{\left(\bar{z}_{k|K}^{(\ell+1,m)} - \bar{z}_{k-1|K}^{(\ell+1,m)}\right)^2 + \sigma_{k|K}^{(\ell+1,m)} + \sigma_{k-1|K}^{(\ell+1,m)} - 2\sigma_{k|K}^{(\ell+1,m)}s_{k-1}^{(\ell+1,m)} + 2\beta}{1+2(\alpha+1)}. \quad (17)$$

For each $\ell$ in the outer EM iteration, the inner iteration over $m$ is repeated until convergence, to obtain the updated values of $\{z_k^{(\ell+1)}\}_{k=1}^K$ and $\{\eta_k^{(\ell+1)}\}_{k=1}^K$ to be passed to the outer EM iteration.

The updated estimate of the Bernoulli success probability at window $k$ and iteration $\ell+1$ is given by $q_k^{(\ell+1)} = \text{logit}^{-1}\big(z_k^{(\ell+1)}\big)$. Starting with an initial guess of the parameters, the outer EM algorithm alternates between finding the expectation of $\{n_{k,r}\}_{k=1,r=1}^{K,R}$ and estimating the parameters $\kappa_1$, $\kappa_2$, $\{z_k\}_{k=1}^{K}$ and $\{\eta_k\}_{k=1}^{K}$ until convergence. Confidence intervals for $q_k^{(\ell)}$ can be obtained by mapping the Gaussian confidence intervals for the Gaussian variable $z_k^{(\ell)}$ via the inverse logit mapping. In summary, the decoder inputs the MEG observations and the envelopes of the two speech streams, and outputs the Bernoulli success probability sequence corresponding to attention to speaker 1.

## 3 Results

### 3.1 Simulated Experiments

We first evaluated the proposed state-space model and estimation procedure on simulated MEG data. For a sampling rate of $F_s = 200$Hz, a window length of $W = 50$ samples (250ms), and a total observation time of $T = 12000$ samples (60s), the binary sequence $\{n_{k,r}\}_{k=1,r=1}^{240,3}$ is generated as realizations of a Bernoulli process with success probability $q_k = 0.95$ or $0.05$, corresponding to attention to speaker one or two, respectively. Using a TRF template of length $0.5$s estimated from real data, we generated 3 trials with an SNR of 10dB. Each trial includes three attentional switches occurring every 15 seconds. The hyper-parameters $\alpha$ and $\beta$ for the inverse-Gamma prior on the state variance are chosen as $\alpha = 2.01$ and $\beta = 2$. This choice of $\alpha$ close to 2 results in a non-informative prior, as the variance of the prior is given by $\beta^2/[(\alpha - 1)^2(\alpha - 2)] \approx 400$, while the mean is given by $\beta/(\alpha - 1) \approx 2$. The mean of the prior is chosen large enough so that the state transition from $q_k = 0.99$ to $q_{k+1} = 0.01$ lies in the $98\%$ confidence interval around the state innovation variable $w_k$ (See Eq. (12)). The hyper-parameters for the von Mises distribution are chosen as $d = \frac{7}{2}KR$ and $c_0 = 0.15$, as the average observed correlation between the MEG data and the model prediction is $\approx$ in the range of $0.1$–$0.2$. The choice of $d = \frac{7}{2}KR$ gives more weight to the prior than the empirical estimate of $\kappa_i$.

Figure 2–A and 2–B show the simulated MEG signal (black traces) and predictors of attending to speaker one and two (red traces), respectively, at an SNR of 10 dB. Regions highlighted in yellow in panels A and B indicate the attention of the listener to either of the two speakers. Estimated values of $\{q_k\}_{k=1}^{240}$ (green trace) and the corresponding confidence intervals (green hull) are shown in Figure 2–C. The estimated $q_k$ values reliably track the attentional modulation, and the transitions are captured with high accuracy. MEG data recorded from the brain is usually contaminated with environmental noise as well as nuisance sources of neural activity, which can considerably decrease the SNR of the measured signal. In order to test the robustness of the decoder with respect to observation noise, we repeated the above simulation with SNR values of 0 dB, $-10$ dB and $-20$ dB. As Figure 2–D shows, the dynamic denoising feature of the proposed state-space model results in a desirable decoding performance for SNR values as low as $-20$ dB. The confidence intervals and the estimated transition width widen gracefully as the SNR decreases. Finally, we test the tracking performance of the decoder with respect to the frequency of the attentional switches. From subjective experience, attentional switches occur over a time scale of few seconds. We repeated the above simulation for SNR $= 10$ dB with 14 attentional switches equally spaced during the 60s trial. Figure 2–E shows the corresponding estimate values of $\{q_k\}$, which reliably tracks the 14 attentional switches during the observation period.

### 3.2 Application to Real MEG Data

We evaluated our proposed state-space model and decoder on real MEG data recorded from two human subjects listening to a speech mixture from a male and a female speaker under different attentional conditions. The experimental methods were approved by the Institutional Review Board (IRB) at the authors' home institution. Two normal-hearing right-handed young adults participated in this experiment. Listeners selectively listened to one of the two competing speakers of opposite gender, mixed into a single acoustic channel with equal density. The stimuli consisted of 4 segments from the book *A Child History of England* by Charles Dickens, narrated by two different readers (one male and one female). Three different mixtures, each 60s long, were generated and used in different experimental conditions to prevent reduction in attentional focus of the listeners, as opposed to listening to a single mixture repeatedly over the entire experiment. All stimuli were delivered

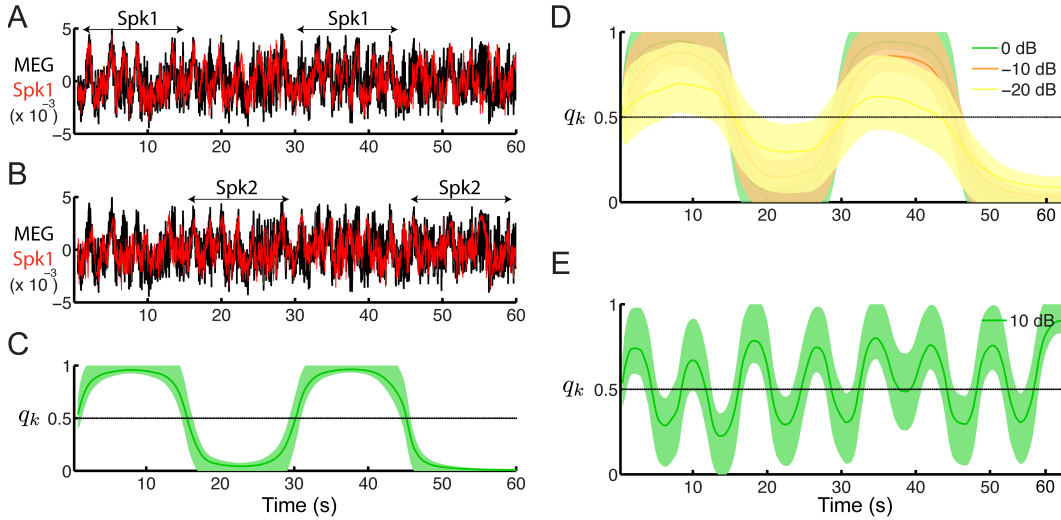

Figure 2: Simulated MEG data (black traces) and model prediction (red traces) of A) speaker one and B) speaker two at $\mathsf{SNR} = 10$ dB. Regions highlighted in yellow indicate the attention of the listener to each of the speakers. C) Estimated values of $\{q_k\}$ with $95\%$ confidence intervals. D) Estimated values of $\{q_k\}$ from simulated MEG data vs. $\mathsf{SNR} = 0, -10$ and $-20$dB. E) Estimated values of $\{q_k\}$ from simulated MEG data with $\mathsf{SNR} = 10$dB and 14 equally spaced attention switches during the entire trial. Error hulls indicate $95\%$ confidence intervals. The MEG units are in $pT/m$.

identically to both ears using tube phones plugged into the ears and at a comfortable loudness level of around 65 dB. The neuromagnetic signal was recorded using a 157–channel, whole-head MEG system (KIT) in a magnetically shielded room, with a sampling rate of 1kHz. Three reference channels were used to measure and cancel the environmental magnetic field [19].

The stimulus-irrelevant neural activity was removed using the DSS algorithm [11]. The recorded neural response during each 60s was high-pass filtered at 1 Hz and downsampled to 200 Hz before submission to the DSS analysis. Only the first component of the DSS decomposition was used in the analysis [9]. The TRF corresponding to the attended speaker was estimated from a pilot condition where only a single speech stream was presented to the subject, using 3 repeated trials (See Section 2.1). The TRF corresponding to the unattended speaker was approximated by truncating the attended TRF beyond a lag of 90ms, on the grounds of the recent findings of Ding and Simon [8] which show that the components of the unattended TRF are significantly suppressed beyond the M50 evoked field. In the following analysis, trials with poor correlation values between the MEG data and the model prediction were removed by testing for the hypothesis of uncorrelatedness using the Fisher transformation at a confidence level of $95\%$ [20], resulting in rejection of about $26\%$ of the trials. All the hyper-parameters are equal to those used for the simulation studies (See Section 3.1).

In the first and second conditions, subjects were asked to attend to the male and female speakers, respectively, during the entire trial. Figure 3–A and 3–B show the MEG data and the predicted $q_k$ values for averaged as well as single trials for both subjects. Confidence intervals are shown by the shaded hulls for the averaged trial estimate in each condition. The decoding results indicate that the decoder reliably recovers the attention modulation in both conditions, by estimating $\{q_k\}$ close to 1 and 0 for the first and second conditions, respectively. For the third and fourth conditions, subjects were instructed to switch their attention in the middle of each trial, from the male to the female speaker (third condition) and from the female to the male speaker (fourth condition). Switching times were cued by inserting a 2s pause starting at 28s in each trial. Figures 3–C and 3–D show the MEG data and the predicted $q_k$ values for averaged and single trials corresponding to the third and fourth conditions, respectively. Dashed vertical lines show the start of the 2s pause before attentional switch. Using multiple trials, the decoder is able to capture the attentional switch occurring roughly halfway through the trial. The decoding of individual trials suggest that the exact switching time is not consistent across different trials, as the attentional switch may occur slightly earlier or later than the presented cue due to inter-trial variability. Moreover, the decoding results for a correlation-based classifier is shown in the third panel of each figure for one of the subjects. At each time window, the

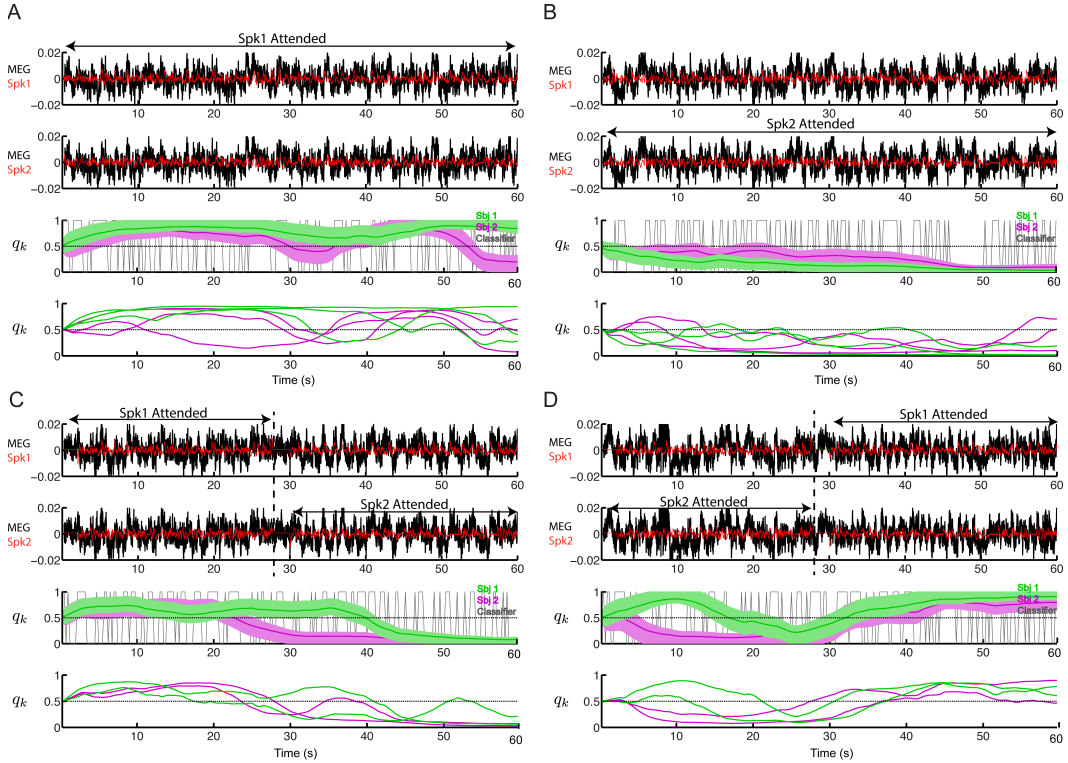

Figure 3: Decoding of auditory attentional modulation from real MEG data. In each subplot, the MEG data (black traces) and the model prediction (red traces) for attending to speaker 1 (male) and speaker 2 (female) are shown in the first and second panels, respectively, for subject 1. The third panel shows the estimated values of $\{q_k\}$ and the corresponding confidence intervals using multiple trials for both subjects. The gray traces show the results for a correlation-based classifier for subject 1. The fourth panel shows the estimated $\{q_k\}$ values for single trials. A) Condition one: attending to speaker 1 through the entire trial. B) Condition two: attending to speaker 2 through the entire trial. C) Condition three: attending to speaker 1 until $t = 28s$ and switching attention to speaker 2 starting at $t = 30s$. D) Condition four: attending to speaker 2 until $t = 28s$ and switching attention to speaker 1 starting at $t = 30s$. Dashed lines in subplots C and D indicate the start of the 2s silence cue for attentional switch. Error hulls indicate $95\%$ confidence intervals. The MEG units are in $pT/m$.

classifier picks the speaker with the maximum correlation (averaged across trials) between the MEG data and its predicted value based on the envelopes. Our proposed method significantly outperforms the correlation-based classifier which is unable to consistently track the attentional modulation of the listener over time.

## 4 Discussion

In this paper, we presented a behaviorally inspired state-space model and an estimation framework for decoding the attentional state of a listener in a competing-speaker environment. The estimation framework takes advantage of the temporal continuity in the attentional state, resulting in a decoding performance with high accuracy and high temporal resolution. Parameter estimation is carried out using the EM algorithm, which at its heart ties to the efficient computation of the Bernoulli process smoothing, resulting in a very low overall computational complexity. We illustrate the performance of our technique on simulated and real MEG data from human subjects. The proposed approach benefits from the inherent model-based dynamic denosing of the underlying state-space model, and is able to reliably decode the attentional state under very low SNR conditions. Future work includes generalization of the proposed model to more realistic and complex auditory environments with more diverse sources such as mixtures of speech, music and structured background noise. Adapting the proposed model and estimation framework to EEG is also under study.

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
