[Reviews · NeurIPS 2014]

Submitted by Assigned_Reviewer_27

In the last few years, there has been considerable interest in the neuroscience community in decoding high-level latent brain states from noisy neural recordings. One such state variable is attention: primates and other high-level organisms can preferentially distribute resources to encode and process a selective set of incoming stimuli in a way that is typically not externally visible. In the case of auditory selective attention, empirical studies have identified a set of neural variables that can be measured that provide information about which sound a subject is currently directing attention to. Some of these variables can be measured via magnetoencephalography (MEG). This study capitalises on these prior observations to build a statistically-principled decoder of human auditory attentional states under competing-speaker auditory stimulation. The authors explain the underlying measurements, represent the problem as one of Bayesian inference, then present an EM-based inference procedure for inferring the latent variables. The authors demonstrate the performance of this algorithm on simulated data, then test it on real MEG data where subjects were directed to attend in a specified way.

At a high level, this is a well-motivated study on a topical problem, and the authors have executed it well.

Quality: As far as I can tell, the inference procedure is technically sound. The authors demonstrate its application to synthetic data where ground truth is known, and they are able to recover the ground truth reasonably well. The results on real data are reasonably convincing, insofar as these are very limited and noisy measurements. It would be useful if the authors added a comment on how to relate the SNRs in the synthetic data with the real data, given that the noise in the real MEG signals (Fig 3, top two rows of each panel) are rather high.

Clarity: Generally, this is well explained, though there were a few points in the text which could have been better motivated :-

- There's some confusion on spectrotemporal vs temporal features of speech streams in this paper. It is true that some previous work uses spectrotemporal information from the speech stream (e.g. Mesgarani & Chang, Ding and Simon). The model presented in this paper only makes use of temporal information. The authors do make this clear initially. But Line 49 refers to Fig 1A as being about the spectrotemporal information (while the image only shows the temporal information). On Line 71, the authors wonder whether one can decode using a "more parsimonious set of spectrotemporal features", but really they are asking whether they can rely on temporal features alone. On Line 94, the authors claim they will decode the attentional state given the spectrotemporal features of the two speech streams, but only do so in terms of the temporal speech streams.
- DSS: I'm not an expert on this denoising technique, but describing the method as "removing inconsistent temporal components that are not phase-locked to the stimulus" sounds an awful lot like it would suppress markers of attentional shifts (which are not typically stimulus-locked). I suspect that this is just a poor choice of wording.
- Section 2.2.1. It really wasn't clear to me until much later on that this section is describing the likelihood. This is perhaps the most confusing part of the paper. The authors should clarify that that the generative model encapsulated in equation (3) is never inverted directly, but rather tested via the surrogate, theta. The authors do need to explain carefully why they rely on this surrogate measure, and not just, for example, consider $y_{t,r}$ as conditionally Gaussian with mean $e_{i,t}$. My suspicions are either that there are slow gain fluctuations in the neural signal, due to changes in global states such as arousal, as well as stimulus-driven gain changes, which the correlation "detrends", or it introduces a nonlinearity that assists in the temporal smoothing. Whether it is because of these or other reasons, the authors should explain why.
- It was not clear to me why the von Mises (prior/posterior) distributions on theta (e.g. eqn (8)) should be zero-mean/mode. Figure 2B of Ding & Simon 2012 suggest that the correlations (across time) between $e_{i,t}$ and $y_{t,r}$ should be positive for both attended and unattended streams.

Originality: This provides a more rigorous and principled approach to auditory attentional-state decoding than the other papers I have seen on this topic (e.g. by Ding & Simon, Mesgarani & Chang), and would likely be a welcome step forward in this domain. Some references would be appreciated for the statement "Previous studies employ time-averaging..." on Line 50.

Significance: This kind of decoding has a strong practical component for scientists interested in studying attentional states, and engineers developing tools for interacting with the attending brain. There's clearly a fair way to go on this topic still (the performance of this algorithm is ultimately limited), but it's a good step forward.

Some additional comments:

- Line 125. If the authors have room, they should insert a paragraph break before "In a competing-speaker environment."

- Line 110: "It is known that the TRF is a sparse filter". The TRF is not known, but estimated. For future work, the authors may consider a different regularisation scheme -- I suspect that the TRF is better described as temporally localised. The components of Fig 1C after 250ms look very artefactual, and are unrealistic given the functional characteristics of the auditory system. The authors may wish to consider Park & Pillow's ALD as an alternative method for estimating tau.

- Lines 169-171 "The subjective experience... by the listener." This is irrelevant, and should be cut.

- Line 416. It's a bit rich to blame the variability in estimated switching time on the subject. It's certainly possible, but it would be more parsimonious to blame the algorithm. If the authors want to comment on this, they should sample from the individual-trial posteriors $p_{k,r}$ to estimate the strength of this prediction. Also, it appears to be that the model's assumption of slowly-varying attentional state (via the AR1 dynamics on $z_k$) breaks down under the instruction that the subject switch attention during the pause. The authors should note this.

- I suggest changing the Bernoulli success probability symbol from $p_k$ to $q_k$, to disambiguate from the pdf symbol $p$. Also equation (10) should start with $p$
Summary: This is a well-motivated study on a topical problem, and the authors have executed it well. A few minor changes should be made here and there.

Submitted by Assigned_Reviewer_29

This paper develops a method for dynamically decoding MEG data to
determine which of a pair of speakers a listener is attending to. The
method employs a probabilistic time-series model with binary variables
indicating the attentional state. The input to the model are a pair of
angular variables at each time-step that indicate which of two
attentional models are a better match to the data at the current
time-point. The efficacy of the approach is demonstrated on synthetic
and real MEG data.

In general I like the paper and the experiments on real data worked
out quite nicely. However, I think there are some issues with the
presentation of the modelling work, and possibly with the model
itself. It would also have been interesting to unpack the performance
of the model with some additional experimental work. For these
reasons, although I think the work is promising, it is not yet ready
for publication in my opinion.

Major comments

I have a couple of issues with the model.

First, on page 4 the equation for the probability of the parameter set
given the observations (on line 192) appears to be incorrect. The
terms for the prior over the concentration parameters of the von Mises
(\kappa_{1:2}) and the variance parameters for the z_k (\eta_k) which
are included inside the sum over R and K. Since there is one global
prior (for all times and trials) I believe that these should not be
inside the sums? Identical comments apply to subsequent equations,
such as the complete data log posterior on line 205, and to the
parameter updates which follow.

Second, the way of framing the model appears non-standard and here I
would have appreciated a detailed discussion. Normally, the data under
consideration during inference and learning are fixed. Here the
\theta_{ikr} play the role of the data and at first sight it might
appear that these are modelled using a mixture of two von Mises
distributions. However, that's not the case since the n_{kr} are used
to select which of the two possible \thetas are modelled. That is the
model has a choice about whether to model \theta_{1,k,r} or
\theta_{2,k,r} (see line 205). In other words, this does not
correspond to a valid generative model for theta. I can see that this
setup will still result in a sensible behavior as it encourages the
model to set n_k=1 if \theta_1 is closest to zero and n_k=0 if
\theta_2 is closest to zero. However some more discussion of this
point is required, especially whether there are any implications for
the EM framework. In light of this, perhaps a more natural model would
treat theta_{1:2,k,r} as bivariate data which are modelled using a
mixture two bivariate von Mises distributions, one with a high
variance in the \theta_1 dimension and a low variance in the \theta_2
dimension, and the other with a high variance in the \theta_2
direction and a low variance in the \theta_1 direction?

The experimental results are nice, but it would have been interesting
and useful to unpack the results some more. For instance, different
features could have been used such as various broadband envelopes
(Hilbert, square and low-pass filter, optimisation based approaches
etc.) and it would certain have been informative to compare
performance to spectro-temporal features to see if there is any
performance difference at all. It would also be illuminating to
include a simple strawman classifier, like assigning the attentional
state using which ever of the \theta_k are smallest at each point, and
then possibly smoothing the result with a lowpass filter. Finally, an
interesting consistency test would corrupt the real data with additive
noise, or through removing missing sections, and check the model makes
estimates which are consistent with those estimated from the
unmodified data (this provides a useful check on the error-bars too).

Minor comments

Abstract: "The resulting decoder is able to track the attentional
modulation of the listener with multi-second resolution", was this
meant to be 'sub-second resolution' (potentially over multiple
seconds)?

There are lots of ways to recover the speech envelope -- please could
you say how E_t is derived from the broad-band signal?

It would be interesting to discuss methods for learning the TRFs using
the same framework, especially as the procedure for determining the
TRF for the unattended speaker is quite ad hoc in a statistical sense.
Summary: A promising paper that could develop into a very interesting publication once the modelling work has been tidied up and the evaluation have been improved. Currently, it is not quite ready yet.

Submitted by Assigned_Reviewer_38

This paper describes an approach for determining which of two speakers an listener is attending to in an audio stream using magnetoencephalographic (MEG) recordings.

The paper is clear in its exposition and is generally of high quality. The model is well motivated, carefully and completely described, and is tested both on simulated and real data. I was a little disappointed that there was not more real MEG data from more subjects and so no really strong statements could be made about the quality of the model given the listeners' attention, but I understand that getting MEG data is expensive and time-consuming.

I do not feel well-versed enough in the current state of the art in this field to make a strong statement about the originality of this approach, but the authors appear to be significantly extending the work of Ding and Simon (2012) to provide a full probabilistic model of speaker attention based on MEG recordings and the temporal envelopes of the two competing speech streams.

There were a few minor niggles:

In Figures 2 and 3 the yellow highlighting showing the listener's (simulated or actual) attention is difficult to see when the paper is printed (even when printed in color).

At various places in section 2, the authors use a sentence structure along the lines of "When θ is close to 0 (π), the MEG data and its predicted value are highly (poorly) correlated". While it's a space-efficient construction, it's also confusing to read, and I'd recommend minimizing its use.

Summary: A clear, well-written, well-structured and well-motivated paper with some interesting results. The lack of more real-world data to evaluate on is somewhat disappointing, but the ideas seem sound.
Author Feedback
Author rebuttal: We would like to thank the anonymous reviewers for their insightful comments and careful evaluation of our work. In what follows, we address the comments of the reviewers and respond to their questions where appropriate.

Reviewer #27:

We agree with the reviewer regarding the notions of "spectrotemporal" and "temporal" features, and will clarify this issue in the revised manuscript. We will also reword the explanation of the DSS algorithm as pointed out by the reviewer. The DSS algorithm is a blind source separation technique that suppresses the components of the data that are noise-like and enhances those that are consistent across trials, with no knowledge of the stimulus or the timing of the task.

The reviewer is absolutely right about our choice of von Mises statistics, as opposed to working with the Gaussian model of Eq. (5): the von Mises statistics are robust with respect to gain fluctuations, and hence make the model amenable to undesired scaling, a virtue that cannot be attained by the Gaussian model. We will explain the rationale of this choice in the revised manuscript.

The von Mises distribution in our model is one-sided, i.e., 0 <= \theta <= \pi, and is not zero-mean. The mean is a decreasing function of kappa and lies in [0, \pi/2]. However, the mode of the distribution is at zero. It is possible to change the mode by adding an offset theta_0 to the distribution, which can in turn be fixed a priori or estimated from data. This indeed will result in a more versatile model with the cost of a more complicated estimation procedure, but our analysis shows that a one-sided zero-mode von Mises distribution sufficiently accounts for the statistics of the data.

Finally, we will gladly apply the comments of the reviewer regarding rewording, notation (p_k to q_k), and discussion of the variability of the switching times. We would also explore ALD as an alternative estimation technique in a follow-up paper.

Reviewer #29:

We agree with the reviewer about moving the prior terms out of the summation in the expressions for complete and incomplete data likelihoods. By absorbing R and K into the hyper-parameters c_0, d, \alpha and \beta and re-defining them, the parameter update equations will slightly change, while there is no noticeable change in the decoding results. We will make this change in the revised manuscript.

We would like to argue that our model is indeed a generative model for {y_{k,r} } (and in turn { theta_{i,k,r} }) and is a special case of the model suggested by the reviewer. First, the model is a scale mixture of two von Mises distributions, as highlighted by the first term of the incomplete log-likelihood (line 192). The variables {n_{k,r} } are the so-called hidden outcomes, commonly used to describe mixture models in a constructive fashion: at time k and trial r, we flip a coin with a success probability of p_k and if the outcome is 1, we draw a sample MEG data y_{k,r} such that \theta_{1,k,r} := \arccos < y_{k,r}/||y_{k,r}||, e_{1,k}/||e_{1,k}|| > is von Mises distributed with parameter \kappa_1, otherwise we draw a sample MEG data y_{k,r} such that \theta_{2,k,r} := \arccos < y_{k,r}/||y_{k,r}||, e_{2,k}/||e_{2,k}|| > is von Mises distributed with parameter kappa_2. Therefore, this procedure generates a set of y_{k,r} (assuming unit norm) distributed as:

P(y_{k,r} ) = p_k / (\pi I_0(kappa_1)) \exp ( \kappa_1 < y_{k,r}, e_{1,k}/||e_{1,k}|| > )
+ (1-p_k) / (\pi I_0(kappa_2)) \exp (\kappa_2 < y_{k, r}, e_{2,k}/||e_{2,k}|| >)

with respect to the (W-1)-spherical coordinates on a principal plane (\phi_1 = \theta_{i,k,r}, \phi_{2} = \phi_{3} = … = \phi_{W-1} = 0). Such generative models are commonly used in statistics (e.g., Gaussian mixture models). The EM theory for such mixture models treats the hidden outcomes {n_{k,r} } as the auxiliary variables and employs their conditional expectation as a means of assigning \theta_{1,k,r} or \theta_{2,k,r} to each time bin k and each trial r. This results in a tractable framework for estimating the model parameters (e.g., clustering for Gaussian mixture models).

Second, let vM(\theta; \kappa) denote the von Mises density with parameter \kappa. Then, the model suggested by the reviewer can be written as follows:

p( y_{k,r} ) = p_k vM(\theta_{1,k,r}; \kappa_1) vM (\theta_{2,k,r}; \kappa_2)
+ (1-p_k) vM(\theta_{1,k,r}; \kappa_3) vM(\theta_{2,k,r}; \kappa_4).

where \kappa_2 << \kappa_1 and \kappa_3 << \kappa_4. The model suggested by the reviewer is a richer model that possibly captures the binary nature of the attentional state more accurately. When viewed as a generative model for y_{k,r}, our model corresponds to the special case of \kappa_2 = \kappa_3 = 0. The EM treatment can be extended naturally to the model suggested by the reviewer, although with the cost of estimating four scale parameters instead of two.

Our analysis shows that the proposed model sufficiently captures the statistics of the data. Therefore, we would like to retain the proposed model in our NIPS submission, while providing a more detailed discussion on our modeling framework in the revised manuscript, and explore the more general model suggested by the reviewer in a follow-up paper.

Finally, the experiments suggested by the reviewer are very thoughtful and interesting. As for the scope of our NIPS paper, we will include the suggested strawman classifier in the revised manuscript. In our work, the envelope is generated using the Hilbert transform. The other suggested experiments involving various temporal covariates as well as those with corrupted and missing data will be explored in a follow-up paper, in the interest of space.

Reviewer #38:

We will change the color mapping of the figures in the revised manuscript to ensure faithful printing results. Also, we are in the process of collecting more data from several new subjects, and will apply our framework to a richer subject pool in a follow-up paper.